# Design of Broad Stopband Filters Based on Multilayer Electromagnetically Induced Transparency Metamaterial Structures

**DOI:** 10.3390/ma12060841

**Published:** 2019-03-13

**Authors:** Ziyu Liu, Limei Qi, Syed Mohsin Ali Shah, Dandan Sun, Bin Li

**Affiliations:** 1School of Electronic Engineering, Beijing University of Posts and Telecommunications, Beijing 100876, China; lexi1022_lzy@bupt.cn (Z.L.); mohsin.shah32@gmail.com (S.M.A.S.); sdd661@bupt.edu.cn (D.S.); 2Beijing Research Center for Information Technology in Agriculture, Beijing 100097, China

**Keywords:** stopband filter, electromagnetically induced transparency, multi-layer

## Abstract

Broad stopband filters are proposed, based on multilayer electromagnetically induced transparency (EIT) metamaterial structures. The single EIT metamaterial consists of a U-shaped resonator and a strip on a polyimide substrate. The EIT-like spectral feature is firstly utilized to achieve stopband filters by properly coupling two layers of EIT structure. Influences of different rotation angles on the transmission properties of the two-layer EIT structure are investigated. It is found the wider low-transmission band can be obtained for the Transverse Magnetic (TM) polarization when the two EIT metal structures are vertical to each other. Furthermore, the bandwidth of the stopband can be controlled by increasing layers of the EIT structures with the proper architectural design. The design using a coupling effect of multi EIT-like resonances in the metamaterial would provide a new method for broad stopband filters in highly integrated optical circuits.

## 1. Introduction

Electromagnetically induced transparency (EIT) is a spectrally narrow optical transmission window accompanied with extreme dispersion [1]. Within this spectral window, dramatically slowed down photons and orders of magnitude enhanced nonlinearities, can enable the manipulation of light at few-photon power levels [2,3,4,5]. Historically, EIT has been implemented in laser-driven atomic quantum systems [6]. However, limited material choices and stringent requirements to preserve the coherence of excitation pathways in atomic systems have significantly constrained the use of EIT effect. Recent studies have revealed that metamaterials can offer a versatile way to obtain the EIT-like optical responses [7,8]. Metamaterials are artificially engineered subwavelength materials which can be designed to display fascinating physical properties that cannot be easily accessed in natural materials [9,10]. For the EIT metamaterial structures, much of the research effort was so far focused on the EIT-like effect with a single resonance in single-layer metal-dielectric metamaterials [11,12] or in a cylindrical all-dielectric metamaterial with a high Q of 1320 [13]. Once stacked in a multilayered structure, the presence of strong near-field coupling between the meta-atoms caused splitting or coupling of the EIT resonances, and lead to multispectral EIT-like behavior. Alp Artar et al. [1] introduced an approach enabling construction of a scalable metamaterial media to support multispectral plasmon-induced transparency. The composite multilayered media consists of coupled meta-atoms with radiant and subradiant hybridized plasmonic modes interacting through the structural asymmetry. 

Zeng et al. [6] investigated the multiple electromagnetically induced transparencies (EIT)-like in graphene metamaterials consisting of a series of self-assembled graphene Fabry-Pérot (FP) cavities. The observed multiple EIT-like windows can be efficiently tuned in the mid-infrared regime by adjusting the Fermi level in graphene and the separations of FP cavities. Lu et al. [14] proposed that a metal–insulator–metal waveguide-resonator system performs a plasmonic analogue of electromagnetically induced transparency (EIT) in atomic systems, the plasmonic EIT-like response enables the realization of nanoscale bandpass filters with multiple channels by coupling with a series of side-coupled cavities and stub waveguides. 

In this paper, broad stopband filters are proposed based on the multi EIT-like resonance in the multi-layer metamaterial structures. Each EIT metamaterial consists of a U-shaped resonator and a strip on a polyimide substrate which acts as bright and bright plasmonic modes coupling. It is found that the EIT-like spectral feature can be utilized to achieve broad stopband filters by proper design of different layers, and bandwidth of the stopband can be controlled by adding layers of the EIT structure with the proper architectural design. In fact, metamaterials with either electric or magnetic responses are inherent transmission band-stop filters, owing to their transmission dips. However, the bandwidth is always narrow due to the resonant nature of the response. Furthermore, to the best of our knowledge, there is little related work being carried out for the suppression of undesired responses or for the elimination of interfering signals in which wider forbidden bands are required [15]. Therefore, our design, by using a coupling effect of multi EIT-like resonances in the metamaterial would provide a new method for broad stopband filters in the highly integrated optical circuits.

## 2. Design

The side view of the unit cell of the single-layer EIT structure is shown in Figure 1a. It consists of two layers. The top layer is made of silver with the U-shaped resonator and a strip [16]. The second layer is selected as polyimide (PI) with the relative permittivity *ε_d_* = 2.1 and tangential loss tg*δ* = 0.008 [17]. The thicknesses of the metal and dielectric layers are t1 = 20 nm and t = 40 nm, respectively. Permittivity of the silver is described by the Drude mode [18,19,20]:(1)$$\varepsilon_{g}=1-\frac{\omega_{p}^{2}}{\omega^{2}+i\omega_{c}\omega }$$
where, the plasma frequency is ω_p_ = 1.366 × 10^16^ rad/s and the collision frequency is ω_c_ = 3.07 × 10^13^ rad/s [16,18]. The dimensions of the other parameters are P = 400 nm, L1 = 130 nm, L2 = 55 nm, L3 = 130 nm, L4 = 140 nm, d = 35 nm, g = 45 nm, h1 = 80 nm, and h2 = h3 = 135 nm. 

In the simulation, commercial software (CST Microwave Studio) is used, and the frequency domain solver is selected to obtain the transmission S_21_ and the reflection S_11_. We use periodic boundary conditions along *x* and *y* axes for the unit cell. For a plane wave which would be normally incident into the surface of the EIT structure from the –z axis, the input and output ports are denoted by the red arrows in Figure 1a. The distances between the ports and the surfaces of the EIT structure are assigned automatically by the CST software as the open (add space) boundary conditions are used along the z-axis for the unit cell. The electromagnetic wave can be divided into the Transverse Electric (TE) polarization where the electric field is parallel to the *y*-axis and the TM polarization where the electric field is parallel to the x-axis (denoted in Figure 1).

## 3. Simulations and Results

Figure 2a shows the transmission and the reflection spectra of the single-layered EIT structure. The top and the bottom layers are corresponding to the TE and TM polarizations, respectively. The solid and dotted lines denote the transmission and reflection curves, respectively. For the TE polarization, the transmission shows the EIT effect at the peak f2 = 420 THz accompanied by two dips at f1 = 396 THz and f3 = 436 THz, respectively, while for the TM polarization, there is only one dip at f4 = 353 THz. Generally, the EIT phenomenon observed in metamaterials can be realized by the bright-bright mode coupling [21] or the bright-dark mode coupling [22]. Classical analogy of EIT effect in metamaterials was initially observed in arrays of asymmetrically split rings (ASRs). The asymmetric coupling between the two bright modes can excite a high-Q mode formed by counter-propagating currents, i.e., a trapped mode resonance [23]. To further understand the physical mechanism of the single-layer EIT structure in Figure 1, Figure 2b shows the electric field distributions of the two polarizations. For the three frequencies on the top layer of the TE polarization, the electric field in the strip is strongly excited at the first dip f1 = 396 THz, while the U-shaped resonator is weakly excited, which means the strip works as a bright mode at f1 = 396 THz. For the second dip at f3 = 436 THz, the left part of the U-shaped structure couples strongly to the incident light, while the two parts on the top and bottom sides have the opposite electric field distributions, which supports the bright mode as the strip at the first dip [24]. At the resonance peak of f2 = 420 THz, both the U shape and the strip are excited simultaneously due to the resonance detuning, and this results in the reverse induced current or electric field distributions for the strip and the left side of the U shape, which corresponds to the characteristics of the trapped mode resonance [16,23,25,26]. 

As a result, a transparency window appears in the transmission spectra. For the TM polarization, there is only one dip at f4 = 353 THz. Based on the distribution of the electric field on the bottom layer, the radiation of the U-shaped resonator is apparently working as a bright mode [21,27,28].

Figure 3a shows the side view of a two-layer EIT structure by rotating the metal structure of the second layer with a clockwise angle ϕ along the z-axis [15,29,30]. The distance between the two layers is S. For the two-layer EIT structure with S = 0, Figure 3b,c gives the color map of transmission curves as ϕ varying from 0° to 180° for the TE and the TM polarizations, respectively. The red and blue color areas correspond to the high-transmission and low-transmission regions, respectively. It is found that there are several regions of low-transmission bands for the two-layer EIT structure. For the TM polarization, the widest low-transmission band varies from 308 THz to 387 THz at ϕ = 90°, which means the broadest stopband filter could be achieved by rotating the angle with ϕ = 90° for the two single-layer EIT structures. For the same EIT structure with ϕ = 90°, Figure 4a,b shows the color map of transmission curves as the distance S varying from 0 nm to 20 nm, for (a) the TE polarization, and (b) the TM polarization. For the TE polarization, there are always three low-transmission regions and this cannot form the broad stopband properties as S increases from 0 nm to 20 nm. As for the TM polarization, there is, firstly, a broadband low-transmission region when S is smaller than 2 nm, but then it splits into three linear curves, and the distances between them tend to enlarge as S increases. Therefore, increasing the distance between two layers of EIT structure will not obtain a wide stopband filter. 

Secondly, it is very difficult to keep a small gap between two layers both in the fabrication and experiment. In the following, we only consider the multilayer structures with the distance S = 0.

Figure 5b shows the transmission spectrum of the TE and TM polarizations of the stopband properties of the two-layer EIT structure with 90° rotation and distance S = 0 (in Figure 5a). Compared with the single-layer EIT structure in Figure 2, three transmission dips appear for each polarization in the two-layer structure due to the hybridization effect and superposition principle between the two layers. Electric field distributions of the resonances are illustrated in Figure 5c. The top layer shows results of the three dips at f1 = 308 THz, f2 = 393 THz and f3 = 419 THz for the TE polarization, the middle layer shows these of the two peaks at f4 = 338 THz and f5 = 410 THz for the TE polarization. The bottom layer represents the results at f6 = 326 THz, f7 = 358 THz and f8 = 387 THz for the TM polarization. For the TE polarization, the U-shaped resonator of the second layer at f1 = 308 THz is strongly excited, which has the same field distribution as the first layer under TM polarization. At f2 = 393 THz, the strip of the first layer and U-shaped resonator of the second layer are excited simultaneously, but the strip has the stronger field than the U-shaped structure. The field distribution of the first layer is nearly the same as that of the single layer EIT structure at 396THz for TE polarization, as is shown in Figure 2b. At f3 = 419 THz, the U-shaped resonator of the first layer is excited and has the similar characteristic of U-shaped resonator of single-layered EIT structure at 436 THz.

For the transmission peaks at f4 = 338 THz and f5 = 410 THz, similar weak electric field distributions can be seen on the two layers, where the strip on the first layer and the right part of U-shaped resonator on the second layer have inverse parallel electric field distributions like the characteristics of electromagnetically-trapped mode of two resonators in a single-layer EIT structure [16,25]. Therefore, the transmission properties of the two-layer EIT structures for the TE polarization are caused by superposition of the two layers of EIT structures. As for the TM polarization, it is observed that the U-shaped resonator on the first layer is excited at f6 = 326 THz, while the others are excited weakly. At f7 = 358 THz, both the U-shaped resonator on the first layer and the strip on the second layer are excited by an incident wave. However, the field in the strip of the second layer is stronger and contributes more to the resonant frequency f7 = 358 THz. At f8 = 387 THz, the field concentrates on the U-shaped resonator of the second layer, which has the similar field distribution of the first layer under the TE polarization as is shown in Figure 2b. Based on the field distributions at the three dips, the transmission of the TM polarization is also the superposition of each layer.

Based on the two-layer EIT structure, a wide low-transmission band appears due to the EIT superposition, which can be designed as a wide stopband filter. Figure 6 shows the transmission spectra of three types of EIT structures with single layer, two layers and three layers for (a) TE polarization and (b) TM polarization. The dot, dot-dash and solid lines denote the results of the single-layer, two-layer and three-layer structures, respectively. Both for the two polarizations, we can see that as the number of the EIT layers increases, more dips will appear in the stop band, which makes a wider stopband in the transmission spectrum. By focusing on the dips of three different layers, it is easy to see that the dips of the three-layer structure can be seen as the superposition of the single-layer and two-layer EIT structures both for the two polarizations, and the broadest stopband filter is obtained for three-layer structure with the TM polarization. 

In the insert figures in Figure 6, the rotating angle ϕ of each two close layers is 90° for the three-layer EIT structure. To generalize, Figure 7 shows the color map of transmission curves for the three-layer structure with ϕ of each two close layers varying from 0° to 180°. For the TE polarization, there are several regions of narrow low-transmission bands, which is difficult to form a broad stopband filter. For the TM polarization, it is easy to form the broad low-transmission region as ϕ locating between 70° to 140°. However, the widest low-transmission band still appears at ϕ = 90° for the three-layer structure.

By introducing the 4th-layer metamaterial into the three-layer EIT structure, a four-layer EIT structure is constructed in Figure 8. For the four-layer EIT structure, the second, third and the fourth layers are the results of 90°, 180°, 270° rotating of the first layer. The dotted, dot-dash and solid lines denote the results of the single-layer, three-layer and four-layer structures, respectively. For the TE polarization, compared with the three-layer structure, bandwidth of the four-layer structure changes little, but better stopband properties are obtained with a sharp out-of-band rejection on the left side. For the TM polarization, it is also seen that the widest stopband is obtained for the four-layer structure.

A 5th-layer metamaterial is introduced into the four-layer EIT structure to get the wider stopband filter in Figure 8, forming a five-layer EIT structure. The last layer is the 360° rotation of the first layer, which has the same direction as the first one. Figure 9 shows the transmission of the five-layer structure. The solid and dot-dashed lines denote the result of the TE and TM polarizations, respectively. It is seen that a wider band-stop filter is obtained due to the superposition of multi-EIT response. For the TM polarization, the bandwidth is wider with low transmission that is lower than 0.3 in the stopband, while a transmission peak of 0.68 appears at 322 THz for TE polarization.

According to the discussions above, the widest stopband characteristics can be obtained for the five-layer structure at the normal incidence. For the same five-layer structure in Figure 9, Figure 10 shows the color map of the transmission spectra at obliquely incident angles varying from 0° to 80°.

We can see that wide stopband filters can be obtained for the two polarizations in the frequency ranges of 220 THz to 480 THz. However, a narrow stopband always exists for the TE polarization. For the TM polarization, the wider stopband filters always appear at the incident angles below 50°. 

For the five-layer structure, the whole thickness is only 300 nm, therefore a substrate is needed for this ultrathin device in practice. Figure 11 shows the transmission spectra of the five-layer system on the substrate of polyimide (PI) and SiO_2_ for the TM polarization at normal incidence. For the SiO_2_, the relative permittivity ε_d_ = 4.41 and tangential loss tgδ = 0.0004 [31] are used in the simulation. The solid, dotted and dot-dashed lines correspond to the results of thickness of substrate PI h = 0, 30 and 70 um, and it is also seen that the property of broad stopband still exists except for a little frequency shift and increase of magnitude in the stop band for the five-layer structure. If SiO_2_ with h = 10 um is used as the substrate, the stopband on the left side shifts to the lower frequency because the increase of the dielectric constant of the whole structure will lead to the decrease of the frequency of the modes according to the variational principles [32,33]. To fabricate the multilayer metamaterial structure, we can firstly get the single-layer EIT structure on the substrate by using electron-beam lithography [24]. As the resolution of the electron-beam lithography system can be controlled within 10 nm [34], the dimensions of the EIT structure given in Figure 1 can be realized in experiment. The multilayer structure can finally be formed by repeating the procedure of single-layer fabrication several times [35]. 

## 4. Conclusions

Broad stopband filters are firstly proposed based on the multi EIT-like resonance in the multi-layer metamaterial structures. Firstly, the physical mechanism of the single-layer EIT structure is given both for the TE and TM polarizations. Then, by investigating the rotation angle on the transmission properties of the two-layer EIT structure, we found that the wider low-transmission band can be obtained for the TM polarization when the two metal resonant structures are vertical to each other. Furthermore, it is found that the bandwidth of the stopband can be controlled by increasing layers of the EIT structure with the proper architectural design, and the wider stopband filter is obtained due to the superposition of multi-EIT response. By comparing the properties of the two-layer, three-layer, four-layer and five-layer EIT structures, the widest stopband filter is obtained for the five-layer structure. For the same five-layer structure, the color map of the transmission spectra at obliquely incident angles varying from 0° to 80°, is also given, and a wider band always appears for the incident angles below 50° for the TM polarization. As different substrates are introduced for the five-layer EIT structure, the broad stopband still exists except for some frequency shift and any increase of the magnitude in the stop band. The proposed broad stopband filters are promising for suppressing undesired responses or eliminating interfering signals within a wide frequency range. Furthermore, while we focused only on a specific multilayer EIT-like metamaterial in the optical frequency, the general principle can be also extended to the longer wavelengths such as terahertz and microwave frequencies, where it is easier to realize the multilayer structure in the manufacturing process.

## Figures and Tables

**Figure 1 materials-12-00841-f001:**
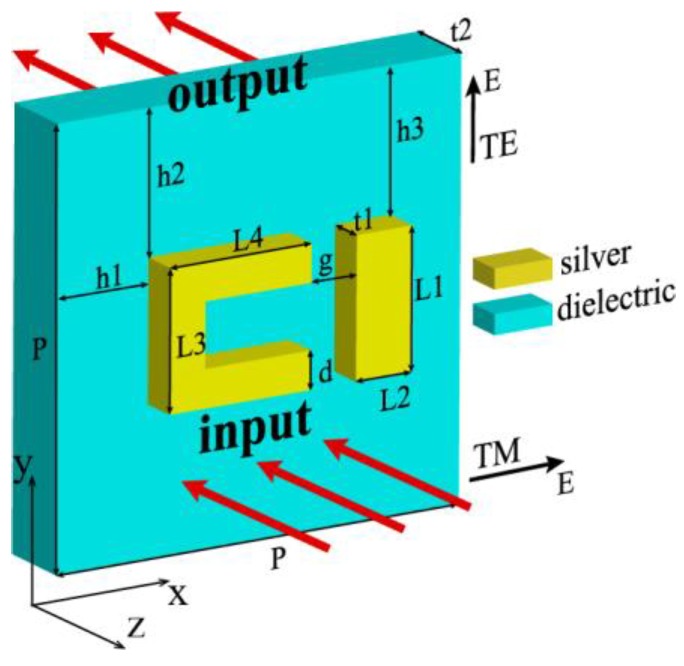
Side view of the unit cell of the single-layer electromagnetically induced transparency (EIT) structure. The dimensions of parameters are P = 400 nm, L1 = 130 nm, L2 = 55 nm, L3 = 130 nm, L4 = 140 nm, d = 35 nm, g = 45 nm, h1 = 80 nm, and h2 = h3 = 135 nm.

**Figure 2 materials-12-00841-f002:**
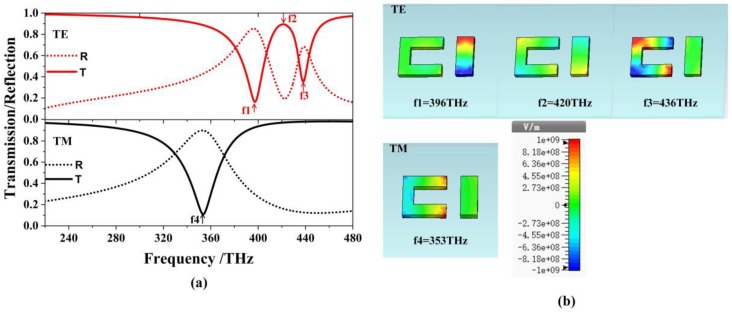
(**a**) The transmission and reflection spectra of the single-layer EIT structure at the normal incidence. (**b**) Electric field distributions of the resonant frequencies at f1 = 396 THz, f2 = 420 THz and f3 = 436 THz for the Transverse Electric (TE) polarization, and f4 = 353 THz for the Transverse Magnetic (TM) polarization.

**Figure 3 materials-12-00841-f003:**
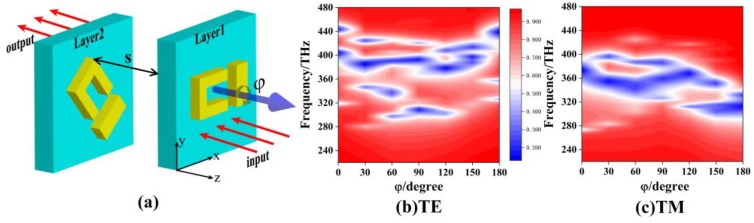
(**a**) The two-layer EIT structure by rotating the metal structure of the second layer with a clockwise angle ϕ along the z-axis. The color map of transmission curves of the two-layer structure with ϕ varying from 0° to 180° for (**b**) the TE polarization and (**c**) the TM polarization.

**Figure 4 materials-12-00841-f004:**
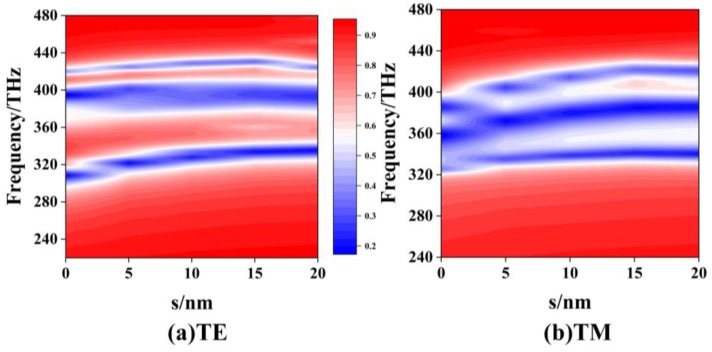
The color map of transmission curves of the two-layer structure with the distance S varying from 0 nm to 20 nm at ϕ = 90° for (**a**) the TE polarization and (**b**) the TM polarization.

**Figure 5 materials-12-00841-f005:**
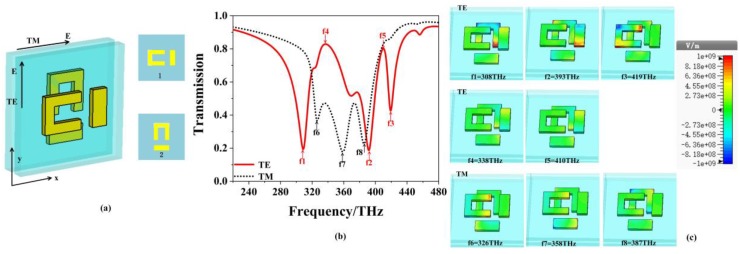
(**a**) Side view of the two-layer EIT structure, where the second EIT structure is formed by rotating the metal structure with 90° from the first structure, (**b**) transmission spectrum of the TE (solid line) and the TM polarizations (dotted line). (**c**) Electric field distributions of eight resonant frequencies for the TE (top layer) and TM polarizations.

**Figure 6 materials-12-00841-f006:**
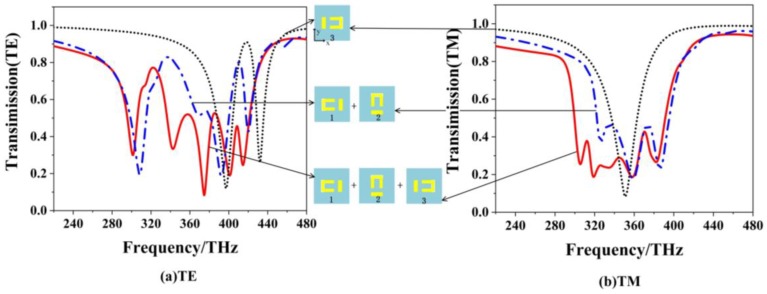
The transmission spectra for three types of EIT structure with single layer, two layers and three layers for (**a**) TE polarization, and (**b**) TM polarization.

**Figure 7 materials-12-00841-f007:**
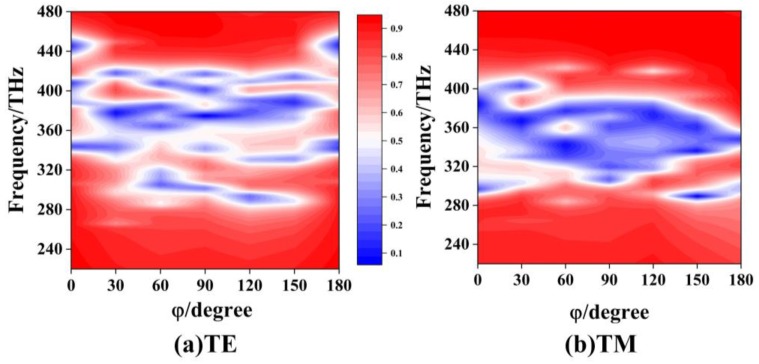
The color map of transmission curves for the three-layer structure with ϕ of each two close layers varying from 0° to 180° for (**a**) the TE polarization and (**b**) the TM polarization.

**Figure 8 materials-12-00841-f008:**
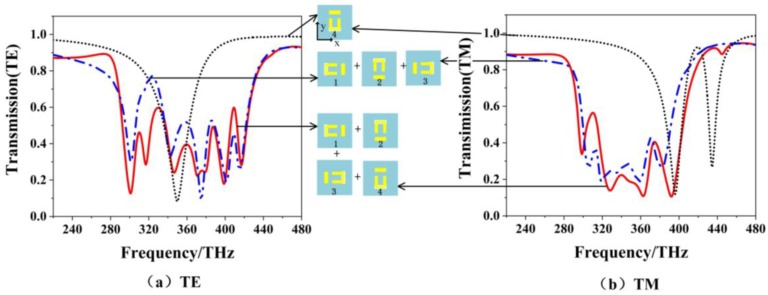
The transmission spectra of three types of EIT structure with single layer, three layers and four layers for (**a**) TE polarization, and (**b**) TM polarization.

**Figure 9 materials-12-00841-f009:**
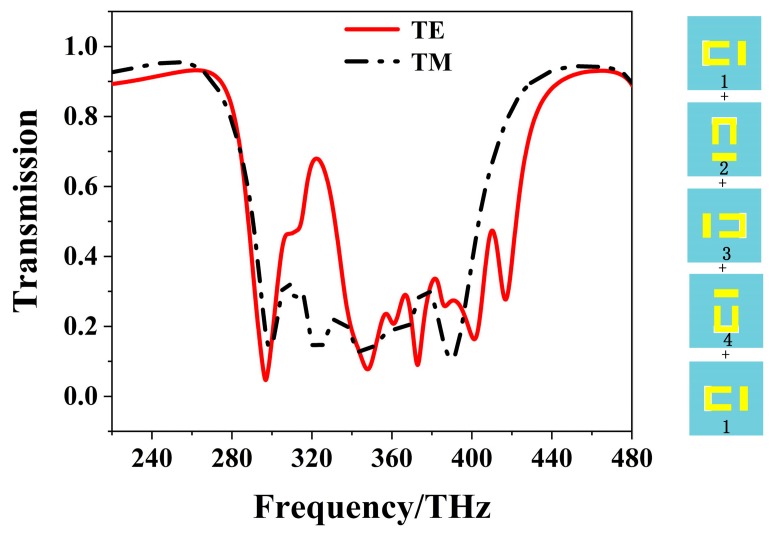
The transmission spectra of five-layer EIT structure for TE polarization and TM polarization.

**Figure 10 materials-12-00841-f010:**
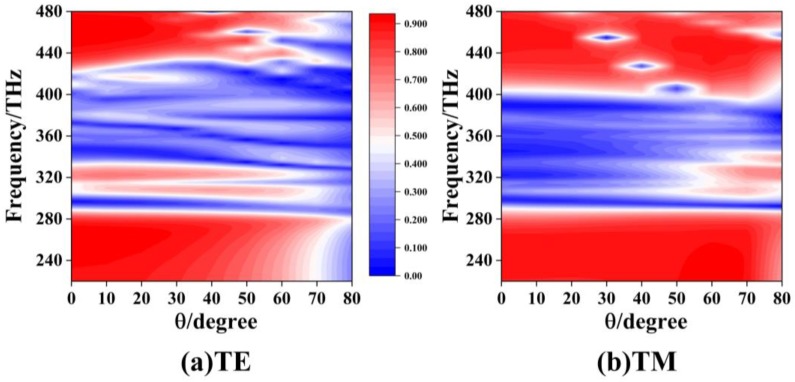
Color map of the transmission spectra for (**a**) the TE and (**b**) the TM polarizations for the five-layer system at obliquely incident angle.

**Figure 11 materials-12-00841-f011:**
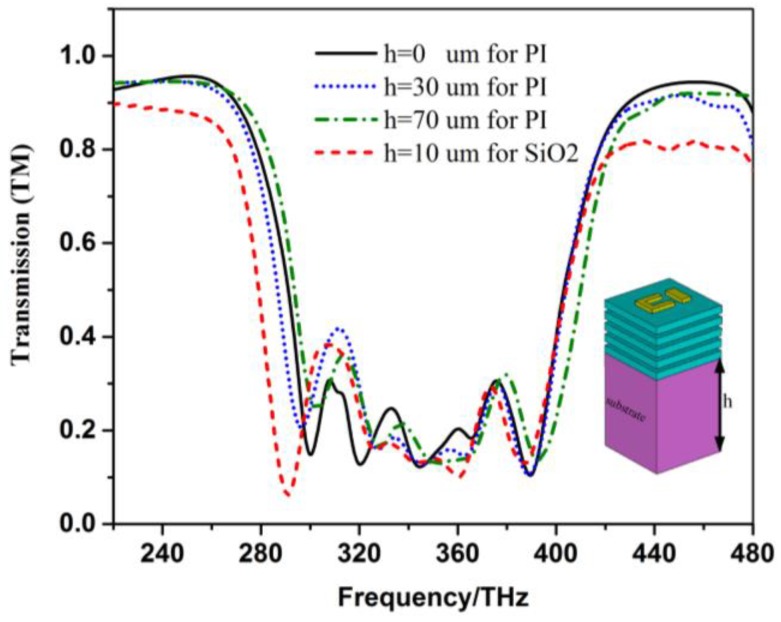
Transmission spectra of the five-layer system on the substrate of PI and SiO2 for the TM polarization at normal incidence.

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
