# Peer review of "Design of Broad Stopband Filters Based on Multilayer Electromagnetically Induced Transparency Metamaterial Structures"

_materials, 2019, doi:10.3390/ma12060841_

Round 1
Reviewer 1 Report
Could you specify the distance between the layers? What is the effects of changing the distance between layers?
How about changing the rotation angle between the layers? For example, in the case of 3 layers, 3 layers could be assembled with rotation angle 120 degree for each other.
Author Response
1. Comments and answers of the review 1:
(1). Could you specify the distance between the layers? What is the effects of changing the distance between layers?
Answer: In the revised manuscript, the distance S between the two layers is denoted in Fig. 3 (a), and the influences of the distance S on the transmission curves are given by the color map in Fig. 4 (the new added figure). From Fig. 3, it is found that the widest stopband filter can be obtained for the TM polarization at ϕ=90°. From Fig. 4, it is found that S=0 is more suitable to get a broad stopband filter when we change the distance S from 0 nm to 20 nm at ϕ=90° both for (a) the TE polarization and (b) the TM polarization,
The corresponding descriptions for Fig. 4 is given in page 4 “For the TM polarization, the widest low-transmission band varies from 308 THz to 387 THz at ϕ=90°, which means the broadest stopband filter could be achieved by rotating the angle with ϕ=90° for the two single-layer EIT structures. For the same EIT structure with ϕ=90°, Fig. 4 (a) and (b) show the color map of transmission curves as the distance S varying from 0 nm to 20 nm for (a) the TE polarization and (b) the TM polarization. For the TE polarization, there are always three low-transmission regions and cannot form the broad stopband properties as S increases from 0 nm to 20 nm. As for the TM polarization, there is, firstly, a broadband low-transmission region when S is smaller than 2 nm, however, it splits into three linear curves and the distances between them tend to enlarge as S increasing. Therefore, increasing the distance between two layers of EIT structure will not obtain a wide stopband filter. Secondly, it is very difficult to keep a small gap between two layers both in the fabrication and experiment. In the following, we only consider the multilayer structures with the distance S=0.”
(2).How about changing the rotation angle between the layers? For example, in the case of 3 layers, 3 layers could be assembled with rotation angle 120 degree for each other.
Answer: We have analyzed the influence of rotation angle on the two-layer EIT structure in Fig. 3. In the revised manuscript, to more generally, Fig. 7 is added to show the color map of transmission curves for the three-layer structure with ϕ of each two close layers varying from 0o to 180o. It is interesting that the widest low-transmission band still appears at ϕ=90° for the three-layer structure with the TM polarization.
The corresponding description for Fig. 7 is given in page 6 “In the insert figures in Fig. 6, the rotating angle ϕ of each two close layers is 90o for the three-layer EIT structure. To generalize, Fig. 7 shows the color map of transmission curves for the three-layer structure with ϕ of each two close layers varying from 0o to 180o. For the TE polarization, there are several regions of narrow low-transmission bands, which is difficult to form a broad stopband filter. For the TM polarization, it is easy to form the broad low-transmission region as ϕ locating between 70o to 140o. However, the widest low-transmission band still appears at ϕ=90° for the three-layer structure.”

Reviewer 2 Report
The paper presents metamaterials-based bandstop filters in terahertz frequencies based on Electromagnetically induced transparency effect. In general interesting. However the electromagnetically induced transparency is not explained clearly. The introduction is missing the recent literature on EIT-like effects such as : Metamaterial Engineered Transparency due to nullifying of multipole moments', Opt. Letters, 43(3), 503-506 (2018). Also the quality factor reported in Opt. Letters, 43(3), 503-506 (2018) is much higher compared to what the present manuscript report on Figure 2.
Author Response
Answer for the questions:
1. In the revised manuscript, more explanations of the electromagnetically induced transparency phenomenon are added in the first paragraph of page 3, it is “Generally, the EIT phenomenon observed in metamaterials can be realized by the bright-bright mode coupling [21] or the bright-dark mode coupling [22]. Classical analogy of EIT effect in metamaterials was initially observed in arrays of asymmetrically split rings (ASRs). The asymmetric coupling between the two bright modes can excite a high-Q mode formed by counter-propagating currents, i.e. a trapped mode resonance [23].”
To shows the physical mechanism of the EIT phenomenon for the single-layer EIT structure in Fig. 1, we find it is the bright-bright mode coupling through the electric field distributions of the three frequencies for the TE polarization. The corresponding description is “For the three frequencies on the top layer of the TE polarization, the electric field in the strip is strongly excited at the first dip f1=396 THz while the U-shaped resonator is weakly excited, which means the strip works as a bright mode at f1=396 THz. For the second dip at f3= 436 THz, The left part of the U-shaped structure couples strongly to the incident light while the two parts on the top and bottom sides have the opposite electric field distributions, which supports the bright mode as the strip at the first dip [24]. At the resonance peak of f2=420 THz, both the U shape and the strip are excited simultaneously due to the resonance detuning, and result in the reverse induced current or electric field distributions for the strip and the left side of the U shape, which corresponds to the characteristics of the trapped mode resonance [16, 23, 25, 26]. As a result, a transparency window appears in the transmission spectra.”
2. In the revised manuscript, three recent literatures on EIT-like effect are cited in the first part of the introduction including Opt. Letters, 43(3), 503-506 (2018). It is “For the EIT metamaterial structures, much of the research effort was so far focused on the EIT-like effect with a single resonance in single-layer metal-dielectric metamaterials [11, 12] or in a cylindrical all-dielectric metamaterial with a high Q of 1320 [13].”
In the literature (Opt. Letters, 43(3), 503-506 (2018)), the authors proposed a metamaterial transparency effect due to the nullifying of the main excited dipole moments, leading to almost zero radiative losses in all dielectric metamaterials ,which is a new idea. However, in the revised manuscript, we focus on the multi EIT-like resonances in metamaterials, the cited papers that showing a single resonance EIT-like effect is not extended.
Reviewer 3 Report
The authors present a study of creating bandstop filters through multilayer metamaterial structures. They clearly show the buildup of the stop band with a judicious increase in layer concatenation. This work is novel, timely and merits publications.
I have only one comment to make:
It would be good for the authors to include a brief statement about the effects of including a substrate for their multilayer metamaterial. This would serve as a potential link from theory to experiment for the readers.
Author Response
Answer: In the last paragraph of Sect. 3, Fig. 11 shows the transmission spectra of the five-layer system on the substrate of polyimide (PI) and SiO2 for the TM polarization at normal incidence. We find that as different substrates are introduced for the five-layer EIT structure, the broad stopband still exists except for some frequency shift and increase of the magnitude in the stopband. The result is very helpful in practice and experiment. In the simulation, only the thickness of the substrate below 100 um is investigated because it will take a few weeks to get the results for the thicker substrate with our computer.
The added paragraph in page 8 is “For the five-layer structure, the whole thickness is only 300 nm, a substrate is needed for this ultrathin device in practice. Fig. 11 shows the transmission spectra of the five-layer system on the substrate of polyimide (PI) and SiO2 for the TM polarization at normal incidence. For the SiO2, the relative permittivity εd =4.41 and tangential loss tgδ=0.0004 [32] are used in the simulation. The solid, dotted and dot-dashed lines are corresponding to the results of thickness of substrate PI h=0, 30 and 70 um, it is seen that the property of broad stopband still exists except for a little frequency shift and increase of magnitude in the stop band for the five-layer structure. If SiO2 with h=10 um is used as the substrate, the stopband on the left side shifts to the lower frequency because the increase of the dielectric constant of the whole structure will lead to the decrease of the frequency of the modes according to the variational principles [33, 34].”
Reviewer 4 Report
Authors should add a theoretical part of the structure design in order to understand the frequency work of different modes and especially the resonant frequencies ( to confirm the filter geometry response as stopband or passband ). In this study all based results are performed by simulation. In my opinion, it is very important to add the theoretical part. Those adjustments, would further increase the clarity and the scientific quality of this article. It would be important to add in conclusion the perspective of uses of the proposed geometries? as for example the future processes to be used for their manufacture and also the potential applications targeted with this type of topologies. 1)The second layer is 64 polymer with the relative permittivity εd =2.1 and tangential loss tgδ=0.008 [13] What is the name of the presented polymer? 2) In the simulation, commercial software 70 (CST Microwave Studio) is used, and the frequency domain solver is selected to obtain the 71 transmission S21 and the reflection S11. Is it possible to show in the geometry the position of the 2 ports accesses? Difficult to see the input and output ports. 3) Figure 2. (a) The transmission spectra of the single-layer EIT structure at the normal incidence. The 78 TE polarization is indicated by a red solid line, while the black dotted line indicates the TM 79 polarization. (b) Electric fields distributions of the resonances frequencies at f1= 396 THz, f2=420 THz 80 and f3=436 THz for the TE polarization, and f4=353 THz for the TM polarization In this figure, authors presented the transmission response (S21) versus frequency. I suggest to add also reflexion reponse (S11 or S22) because this spectra will help to confirm the different resonant frequencies. I think that F1 and F3 are not a resonant frequencies and the resonator has just F2 as resonant frequency because of the good repartition of electric field as presented on figure 2.b! I think that F1 and F3 present the minimum of transmission as we can see it in figure 2.b. 4) Figure 4. (a) Side view of the two-layer EIT structure, where the second EIT structure is formed by 115 rotating the metal structure with 90o from the first structure, (b) transmission spectrum of the TE 116 (solid line) and the TM polarizations (dotted line). (c) Electric field distributions of six resonant 117 frequencies for the TE (top layer) and TM polarizations. Why authors didn’t represent the electric field repartition around frequencies 340 and 410THz because the transmission amplitude seems maximum?
Author Response
Answers:
1. In the revised manuscript, more explanations of the electromagnetically induced transparency phenomenon are added in the first paragraph of page 3, it is “Generally, the EIT phenomenon observed in metamaterials can be realized by the bright-bright mode coupling [21] or the bright-dark mode coupling [22]. Classical analogy of EIT effect in metamaterials was initially observed in arrays of asymmetrically split rings (ASRs). The asymmetric coupling between the two bright modes can excite a high-Q mode formed by counter-propagating currents, i.e. a trapped mode resonance [23].”
To shows the physical mechanism of the EIT phenomenon for the single-layer EIT structure in Fig. 1, we find it is the bright-bright mode coupling through the electric field distributions of the three frequencies for the TE polarization. The corresponding description is “For the three frequencies on the top layer of the TE polarization, the electric field in the strip is strongly excited at the first dip f1=396 THz while the U-shaped resonator is weakly excited, which means the strip works as a bright mode at f1=396 THz. For the second dip at f3= 436 THz, The left part of the U-shaped structure couples strongly to the incident light while the two parts on the top and bottom sides have the opposite electric field distributions, which supports the bright mode as the strip at the first dip [24]. At the resonance peak of f2=420 THz, both the U shape and the strip are excited simultaneously due to the resonance detuning, and result in the reverse induced current or electric field distributions for the strip and the left side of the U shape, which corresponds to the characteristics of the trapped mode resonance [16, 23, 25, 26]. As a result, a transparency window appears in the transmission spectra.”
2. Just as the reviewer 4 pointed out, it is more important to add the theoretical part. Some paper once used the LC theory model to analyze the EIT effect based on the simulation. However, we find it is really difficult to theoretically analyze the model without any simulation in the limited time. In the long run, we will keep on this theory work and try to find some useful methods.
3. In the revised manuscript, at the end paragraph of the introduction part, we show why we investigate the broad stopband filter as well as its potential application. The added part is “In fact, metamaterials with either electric or magnetic responses are inherent transmission band-stop filters owing to their transmission dips. However, the bandwidth is always narrow due to the resonant nature of the response. Furthermore, to the best of our knowledge, there is little related work being carried out for the suppression of undesired responses or for the elimination of interfering signals in which wider forbidden bands are required [15]. Therefore, our design, by using coupling effect of multi EIT-like resonances in the metamaterial would provide a new method for broad stopband filters in the highly integrated optical circuits.”
(1).The second layer is polymer with the relative permittivity εd =2.1 and tangential loss tgδ=0.008 [13] What is the name of the presented polymer?
Answer: In the revised manuscript, Polyimide (PI) is given for the present polymer as it is widely used in terahertz and optical bands, the permittivity can vary from 1.8 to 3.1.
(2).In the simulation, commercial software (CST Microwave Studio) is used, and the frequency domain solver is selected to obtain the transmission S21 and the reflection S11. Is it possible to show in the geometry the position of the 2 ports accesses? Difficult to see the input and output ports.
Answer: In Fig. 1 (a), the input and output ports are denoted by red arrows. The corresponding description for Fig. 1(a) is added in the first paragraph of page 3. It is “For a plane wave normally incident into the surface of the EIT structure from the –z axis, the input and output ports are denoted by the red arrows in Fig. 1 (a). The distances between the ports and the surfaces of the EIT structure is assigned automatically by the CST software as the open (add space) boundary conditions are used along the z axis for the unit cell.” Besides, to make it easy to understand, the location between the two layer structures in Fig. 3 (a) is changed, and the input and output ports are also denoted as well.
(3). Figure 2. (a) The transmission spectra of the single-layer EIT structure at the normal incidence. The TE polarization is indicated by a red solid line, while the black dotted line indicates the TM polarization. (b) Electric fields distributions of the resonances frequencies at f1= 396 THz, f2=420 THz and f3=436 THz for the TE polarization, and f4=353 THz for the TM polarization. In this figure, authors presented the transmission response (S21) versus frequency. I suggest to add also reflexion response (S11 or S22) because this spectra will help to confirm the different resonant frequencies. I think that F1 and F3 are not a resonant frequencies and the resonator has just F2 as resonant frequency because of the good repartition of electric field as presented on figure 2.b! I think that F1 and F3 present the minimum of transmission as we can see it in figure 2.b.
Answer: In the revised manuscript, the reflection responses are added both for the TE and the TM polarization in Fig. 2 (a). The corresponding description is “Fig. 2(a) shows the transmission and the reflection spectra of the single-layered EIT structure. The top and the bottom layers are corresponding to the TE and TM polarizations, respectively. The solid and dotted lines denote the transmission and reflection curves, respectively.”
There is no doubt that F2 is a resonant frequency. To show whether F1 and F3 are the resonant frequencies or not, Fig. S shows the transmission curves of the single layer EIT structure with U+strip (solid), only U (red dot) and only strip (blue dot) structures. It is found that transmission dips of the U+strip structure are just the coupling of the U and stipe structure. In Fig. S, electric field distributions of the single U and strip structure are also shown, which are corresponding to the results of the U+strip structure in Fig. 2(b) at the two dips of F3 and F1, respectively. Therefore, we can confirm that the F1 and f3 are also the resonant frequencies of the U+strip structure.
(For the figure,please see the attachement )
Fig. S. Transmission curves of the single layer EIT structure with both the U+strip (solid), only U (red dot) and strip (blue dot) structures (left). The electric field distributions of the single U and single strip structure (right).
(4). Figure 4. (a) Side view of the two-layer EIT structure, where the second EIT structure is formed by rotating the metal structure with 90o from the first structure, (b) transmission spectrum of the TE (solid line) and the TM polarizations (dotted line). (c) Electric field distributions of six resonant frequencies for the TE (top layer) and TM polarizations. Why authors didn’t represent the electric field repartition around frequencies 340 and 410THz because the transmission amplitude seems maximum?
Answer: In the revised manuscript, the two transmission peaks at 338 and 410THz for the TE polarization are denoted in Fig. 5 (b), and their electric fields are also shown in the middle layer of Fig. 5 (c). It is found that the electric distributions of the two peaks like the characteristics of electromagnetically-trapped mode of two resonators in a single-layer EIT structure.
The descriptions for Fig. 5(c) are given in the first and second paragraphs of page 5. The first part is “The top layer shows results of the three dips at f1=308 THz, f2=393 THz and f3=419 THz for the TE polarization, the middle layer shows these of the two peaks at f4=338 THz and f5=410 THz for the TE polarization. The bottom layer represents the results at f6=326 THz, f7=358 THz and f8=387 THz for the TM polarization.”
The second part is “For the transmission peaks at f4=338 THz and f5=410 THz, similar weak electric field distributions can be seen on the two layers, where the strip on the first layer and the right part of U-shaped resonator on the second layer have inverse parallel electric field distributions like the characteristics of electromagnetically-trapped mode of two resonators in a single-layer EIT structure [16, 25]. Therefore, the transmission properties of the two-layer EIT structures for the TE polarization are cause by superposition of the two layers of EIT structures.”

Reviewer 5 Report
In the submitted manuscript entitled “Broad Bandstop Filters Based on Multilayer Electromagnetically Induced Transparency Metamaterial Structures,” the author proposed broadband filter realized using multilayered structures with U-shaped resonator. Although paper contains detailed analysis of the multilayer configuration step-by-step for different polarization, the novelty of the paper is low and the experimental verification of the proposed results is missing. The proposed results also show that the proposed topology due to the asymmetry does not have the same responses for different polarization. On the other hand, the authors use some term that are not specific in this field. For these reasons, I consider that the manuscript is not suitable for publication and should be rejected.
Author Response
Answer: Thank you for the suggestion. Now, the revised manuscript has been greatly improved according to the comments of four reviewers. For the manuscript, there is little novelty for the design of single-layer EIT structure. However, to the best of our knowledge, the novelty is that we firstly propose to design a broad stopband filter based on the multi EIT-like resonance in the metamaterial structures, and bandwidth of the stopband filter can be controlled by adding layers of the EIT structure with the proper architectural design.
In the attachment, you can see all the comments and answers for the revised manuscript.

Round 2
Reviewer 4 Report
The authors have spread to a good part of my questions. However, I would like to ask two questions or even suggestions:
1) Given the dimensions of the simulated structures: for example Figure 1. Side view of the unit cell of the single-layer EIT structure.The dimensions of parameters are P = 400 nm, L1 = 130 nm, L2 = 55 nm, L3 = 130 nm, L4 = 140 nm, d = 35 nm, g = 45 nm, h1 = 80 nm, and h2 = 67.h3 = 135 nm.
How will this simulated structure be realized?did the authors think about manufacturing processing?Because the dimensions of the patterns are nanoscale and the accuracy will be difficult? any Suggestions please?
2) Finally, as the works presented in this article are excluded from the simulation.So I suggest introducing simulation in the title of the article.
For the conclusion, it would also be interesting to add perspectives such as manufacturing and applications.
Author Response
Comments and answers for reviewer 4:
The authors have spread to a good part of my questions. However, I would like to ask two questions or even suggestions:
1) Given the dimensions of the simulated structures: for example Figure 1. Side view of the unit cell of the single-layer EIT structure. The dimensions of parameters are P = 400 nm, L1 = 130 nm, L2 = 55 nm, L3 = 130 nm, L4 = 140 nm, d = 35 nm, g = 45 nm, h1 = 80 nm, and h2 = 67.h3 = 135 nm. How will this simulated structure be realized? did the authors think about manufacturing processing? Because the dimensions of the patterns are nanoscale and the accuracy will be difficult? any Suggestions please?
Answers:
To get the dimensions of the simulated structure in the fabrication, we can use the electron-beam lithography method to get the metal structure without difficulty because the resolution can be controlled within 10 nm for the electron-beam lithography system (Ref. 35 Opt. Express 2010, 18, 17187-17192; Ref. 36 Applied Surface Science 2000, 164, 111–117). For the multilayer structure, we can repeat the single-layer structures to form the multilayer structure (Ref. 37 Nature Materials 2008, 17, 31-37). For the dielectric spacer, we can use the spin coating method.
In the revised manuscript, the corresponding sentences are added in page 8. It is “To fabricate the multilayer metamaterial structure, we can firstly get the single-layer EIT structure on the substrate by using the electron-beam lithography [35]. As the resolution of electron-beam lithography system can be controlled within 10 nm [36], the dimensions of the EIT structure given in Fig. 1 can be realized in experiment. The multilayer structure can finally be formed by repeating the procedure of single-layer fabrication several times [37].”
2) Finally, as the works presented in this article are excluded from the simulation. So I suggest introducing simulation in the title of the article.
For the conclusion, it would also be interesting to add perspectives such as manufacturing and applications.
Answers: In the revised manuscript, the title is changed to “Design of Broad Stopband Filters Based on Multilayer Electromagnetically Induced Transparency Metamaterial Structures.” to show that the works presented in this article are excluded from the simulation, not the fabrication.
At the end of the conclusion, the perspectives for manufacturing and applications are added. It is “The proposed broad stopband filters are promising for suppressing undesired responses or eliminating interfering signals within a wide frequency range. Furthermore, while we focused only on a specific multilayer EIT-like metamaterial in the optical frequency, the general principle can be also extended to the longer wavelengths such as terahertz and microwave frequencies, where it is easier to realize the multilayer structure in the manufacturing process.”
Important revisions are given in the following
1) The title of the manuscript is changed.
2) 3 new references (35-37) are cited and added in the revised manuscript.
3) Some sentences are added in page 8 and page 9 to improve the manuscript.
Reviewer 5 Report
In the resubmitted manuscript, the author proposed broadband filter realized using multilayered structures with U-shaped resonator. Although authors made some improvement, the novelty of the paper is still low and the experimental verification of the proposed results is missing. Furthermore, the advantages of the proposed design in not demonstrated. Comparison with the other solutions in term of bandwidth, fabrication complexity, etc. proposed in the literature has to be added. For these reasons, I consider that the manuscript is not suitable for publication and should be rejected.
Author Response
Thanks for your suggestion. Just as you point out that it is better to fabricate the multi-layer structure and demonstrate it through experiment. However, due to the limitation of time and other reasons, we can not show it now. In the long run, we will focus more on the experiment results based on the simulations.